

# Estimation of ice melt, freshwater budget, and their multi-decadal trends in the Baffin Bay and Labrador Sea

Vigan Mensah[1], Koji Fujita[2], Stephen E. L. Howell[3], Miho Ikeda[4], Mizuki Komatsu[5], and Kay I. Ohshima[1,6]

[1]Institute of Low Temperature Science, Hokkaido University
[2]Graduate school of environmental studies, Nagoya University
[3]Climate Research Division, Environment and Climate Change Canada, Toronto, Canada
[4]Faculty of Science, Hokkaido University
[5]Graduate School of Environmental Science, Hokkaido University, Sapporo, Japan
[6]Arctic Research Center, Hokkaido University, Sapporo, Japan

*Correspondence to*: V. Mensah, vmensah@lowtem.hokudai.ac.jp

**Abstract.** The Labrador Sea and contiguous Baffin Bay play an important role in the formation of the upper layer of the North Atlantic Deep Water, an essential component of the Atlantic Meridional Ocean Circulation. The hydrography of these two seas is strongly influenced by the melting of sea-ice and glacier-ice, which has likely been affected by long-term climate changes. In this study, we use historical data of ocean temperature and salinity from 1950 to 2022 to estimate the summer freshwater volume (SFV) in Baffin Bay and the Labrador Sea, establish climatologies, and assess the impact of multi-decadal climate change. The SFV climatology (1956 km$^3$) and the summer freshwater budget (2286 km$^3$) estimated from various components are in good agreement. Sea ice and glacial melt account for 37% and 26% of the freshwater budget, respectively. SFV climatologies before and after 1995 reveal an increase in Baffin Bay (+226 km$^3$) because of enhanced glacier melting, and a decline (-112 km$^3$) in the Labrador Sea because of recent sea ice volume decreases. The time series of Labrador Sea SFV and total freshwater content are uncorrelated at the multi-decadal scale possibly because the influx of freshwater from the Beaufort Sea dominates the long-term variability.

## 1 Introduction

The Labrador Sea plays a crucial role in the global ocean, as it is one of the formation regions of the North Atlantic Deep Water, whose upper layer is made of Labrador Sea Water (Clarke and Gascard, 1983; Smethie et al., 2000). As such, the circulation, water properties, and stratification in the Labrador Sea contribute to the Atlantic Meridional Overturning Circulation (AMOC) and its temporal variability (e.g., Stouffer et al., 2006; Roberts et al., 2013; Buckley and Marshall, 2016; Thornalley et al., 2018). The Labrador Sea Water is formed by winter convection, as the combined effect of fall and winter storms, strong atmospheric cooling, and a preexisting weak stratification leads to deep convection events, with the mixed layer reaching depths of up to 1600 m (Clarke and Gascard, 1983; Våge et al., 2008). The properties of the surface waters and the associated stratification in the Labrador Sea play an important role in establishing deep convection events, and it is thus essential to document the temporal changes in these waters' properties.

The surface waters in the Labrador Sea are typically cold and fresh (Clarke and Gascard, 1983; Pickart and Spall, 2007), and these properties originate from the large freshwater fluxes in both the Labrador Sea and the contiguous Baffin Bay. Besides their geographical collocation, these two seas are connected through a cyclonic current system starting off the coast



of southern Greenland (Fig. 1). The West Greenland Current flows northward along the coastal regions of western Greenland from the southern tip of the island to about 75°N, beyond which it turns westward and merges with southward currents to form the Baffin Bay Current. This current flows southward along the Baffin Island coast and becomes the Labrador Current south of the Davis Strait (Fig. 1). Through its inshore and offshore branches, the Labrador Current then feeds the Labrador Sea shelf

and interior with its fresh surface waters (Fratantoni and McCartney, 2010; Wu et al., 2012; Jutras et al., 2020; 2023). The cumulation of the freshwater discharge all along this anticlockwise circulation explains the cold and fresh properties of the surface waters in the Labrador Sea.

The main sources of freshwater in Baffin Bay and the Labrador Sea can be divided into "local" and "remote" sources. In summer, the local sources of freshwater are sea ice melt, glacier ice melt, sea ice exported from the Arctic Ocean,

precipitation, and river runoff. Baffin Bay is a major source of sea ice due to the existence of the North Water Polynya (Barber et al., 2001; Münchow et al., 2016). With yearly ice production estimates ranging between 115 km$^3$ and 196 km$^3$ per year (Preußer et al., 2019; Ren et al., 2022, Nakata and Ohshima, 2022), it is the polynya with the highest sea ice production in the Arctic Ocean (Tamura and Ohshima 2011). The sea ice produced in this and other polynyas along the Baffin and Newfoundland coasts drifts mainly southward throughout the winter and spring (Jordan and Neu, 1982; Kwok, 2007; Landy et al., 2017), and

then melts from May to September. Besides sea ice production in local polynyas, sea ice is also exported from the Arctic Ocean to the Baffin Bay through the Nares Strait (Kwok, 2005; Kwok et al., 2010; Münchow et al., 2016; Moore et al., 2021; Howell et al., 2023). This southward transport of sea ice occurs mainly in fall/winter, though transport has also been reported in other months of the year (Kwok et al., 2010; Moore et al., 2021). Significant transport of ice to Baffin Bay also occurs from the Canadian Arctic Archipelago via Lancaster Sound and Jones Sound and Hudson Bay via Hudson Strait (Kwok, 2007; Agnew

et al., 2008; Bi et al., 2019). The presence of numerous glaciers on the western coast of Greenland, as well as Devon, Ellesmere, and Baffin islands, is another major source of freshwater (Valeur et al. 1996; Tang et al., 2004; Williamson et al., 2008). These glaciers release freshwater throughout the year via iceberg calving, and in spring and summer via melting. The precipitation minus evaporation budget is positive in Baffin Bay and the Labrador Sea (Walsh and Portis, 1999; Myers et al., 2007), which results in direct freshwater fluxes into the ocean, and significant river runoff, mainly through the rivers in Newfoundland (Déry

et al., 2016). We define these freshwater sources as local because they exhibit a clear low-salinity signal within the upper ~ 50-100 m following their release in spring and summer. This signal then disappears in fall and winter because of turbulent or convective mixing. Thus, these freshwaters released by local sources with a clear surface signature will be called summer freshwater (thickness, volume, content, or budget) hereafter.

The remote source of freshwater consists of water exported from the Beaufort Sea (e.g., Carmack et al., 2008; Zhang et al.,

2021), the largest freshwater reservoir in the Arctic Ocean. This water is exported through the Nares Strait and other straits of the Canadian Arctic Archipelago and is not expected to exhibit a clear near-surface signature due to the long time and distance required for their transport from the Beaufort Sea to the Labrador Sea. The quantification of this remote source of freshwater is not trivial and has not been carried out yet in this region. Freshwater is also exported to a lesser extent from Hudson Bay (Granskog et al., 2009).



While estimates of some of the individual freshwater budget components exist (e.g., Myers et al., 2007; Déry et al., 2016; Landy et al., 2017), there are only a few comprehensive freshwater budget studies that combine all the freshwater sources in Baffin Bay and the Labrador Sea (Serreze et al., 2006; Haine et al., 2015). These studies rely on the estimation of the freshwater content based on a reference salinity (typically between 34.5 and 34.8). Such freshwater content estimates include both local and remote sources of freshwater and allow for an easy comparison of all budget terms. However, the precise quantification

of freshwater melted from sea ice and other local sources cannot be directly obtained through such a method.

        Over the past few decades, several of the freshwater sources in the Labrador Sea have undergone significant changes because of the effects of global warming and Arctic Amplification. For example, Moore et al. (2021) reported that, over the past two decades, the duration of the ice arches that prevent sea ice from being transported from the Arctic Ocean to Baffin Bay has been reduced. As a result, the sea ice volume flux was 70% higher in 2017-2019 than it was in 1997-2009. The

discharge of ice from West Greenland glaciers has also increased significantly (Mankoff et al., 2020) over the past four decades. Moreover, sea ice area in Baffin Bay has decreased by upwards of 15% per decade since 1968 (Derksen et al., 2018), and sea ice thickness has decreased by ~30% per decade since 1996 (Glissenaar et al., 2023). While several of these changes have been documented, a detailed assessment of the long-term (50-100 years) variations of all the freshwater sources in Baffin Bay and the Labrador Sea has yet to be conducted.

A reason for this is the relatively short record of sea-ice thickness data. In-situ observations of sea ice thickness (e.g., the Unified Sea Ice Thickness Climate Data Record; Lindsay, 2010) are not adequate for estimating a freshwater budget due to their sparse spatial and temporal resolution. Satellite-based sea ice thickness data has been recorded only since 2003 and the various satellites successively deployed (e.g., IceSat-1/2, CryoSat-2, or SMOS) cover different orbits and use different sensors or processing algorithms. While sea-ice thickness datasets obtained by merging various satellite data (e.g., Landy et al., 2017)

or via proxy methods (Glissenaar et al., 2023) cover increasingly longer periods, their time-span is still insufficient to estimate changes in thickness or volume over periods longer than 30 years. Thus, an alternative estimation method should be sought to document long-term changes in summer freshwater content, especially those related to the melting of sea ice. A considerable amount of temperature and salinity profiles have been acquired during the ice melting season (~May to September) since the 1950s, and using these data could be a good way to document long-term changes in the summer freshwater inputs in Baffin

Bay and the Labrador Sea.

In this paper, we introduce a method to estimate the local summer freshwater thickness in the Baffin Bay and Labrador Sea (section 2) and document the distribution of freshwater in this region through climatologies. In section 3, we compare the climatological summer freshwater volume with the summer freshwater budget estimated by the summation of all its local contributions: winter sea ice volume, spring and summer ice transport from the Arctic and Canadian Arctic Archipelago, ice

melt and iceberg calving from glaciers, river runoff, and precipitation minus evaporation (P-E) over the sea. Following this validation, we document long-term changes in the summer freshwater volume (SFV) from 1950 to 2022 in section 4. Lastly, we compare our estimates of SFV with total Freshwater Content (FWC) estimated following Carmack et al. (2008) and provide a discussion on the relative importance of SFV on the total freshwater content.





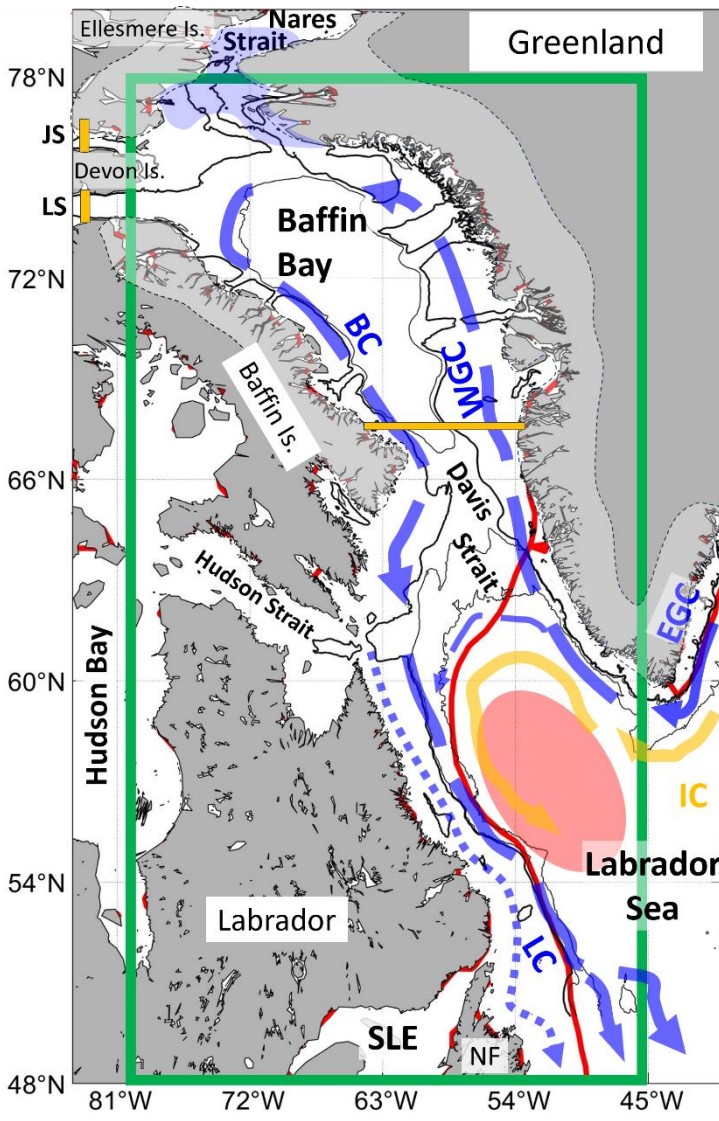

**Fig. 1: Map of Baffin Bay and the Labrador Sea. The green box represents our study region, with the Hudson Bay and the St-Lawrence River estuary being excluded. The red line represents the maximum extent of sea ice in winter. The orange and blue long-dashed lines indicate high-salinity and low-salinity currents, respectively. The initials stand for (SLE) the St-Lawrence River estuary, (NF) Newfoundland, (JS) Jones Sound, (LS) Lancaster Sound, the (IC) Irminger Current, (EGC) East Greenland Current, (WGC) West Greenland Current, (BC) Baffin Current, and (LC) Labrador Current, with the near-shore branch of this current represented by the short-dashed line. The red oval shade represents the region where deep convection associated with the formation of Labrador Sea Water occurs. The white shades on land represent the general areas of marine-terminating glaciers, and the blue shade represents the approximate location of the North Water Polynya. The thick and the two thin solid black lines represent the 500 m, and the 1000 m and 2500 m isobaths, respectively. The straight orange lines represent the gates used to estimate the ice volume flux in the Jones Sound, Lancaster Sound, and Davis Strait. The information in this map was interpreted from Cuny et al. (2002), Rhein et al. (2007), Williamson et al. (2008), Jutras et al. (2020), Mankoff et al. (2020), and Nakata and Ohshima (2022).**





## 2. Data and methods

### 2.1 Freshwater thickness estimation

#### 2.1.1 Temperature and salinity profiles

The main source of temperature and salinity data for this study is historical conductivity-temperature-depth (CTD) data from
the World Ocean Database 2018 (WOD18, Boyer et al., 2018), supplemented with data from the Marine Mammals Exploring the Ocean Pole to Pole (MEOP) database of CTD biologging observations (Treasure et al., 2017). Data were selected from May to August in the Labrador Sea and from May to September in Baffin Bay, that is, up to one month after the end of sea ice melt in the respective regions. Analyzing data from these months ensures that the ice melt or freshwater input signature is clearly visible from the data (section 2.1.2). The study region excludes the St Laurent estuary because the water from this river
outflows south of our study region. A total of 47942 profiles are available, spanning from 1950 to 2022.

#### 2.1.2 Freshwater thickness estimation principle

The basis for our estimation method is the evaluation of the changes in salinity occurring in spring/summer within the upper 100 m of the water column. In fall and winter, the polar ocean's upper layer homogenizes due to the effects of open-ocean or sea ice formation-induced convection. The resulting mixed layer has a temperature close to the freezing point. In the ice-
covered and coastal areas of our study region (Fig. 1), the mixed layer depth may reach up to 100 m (Tang et al., 2004). The cold mixed layer thus formed will be called Winter Water hereafter. When freshwater is released in spring due to glacier or sea ice melting, river runoff, or precipitation, a layer of freshwater forms at the ocean's surface, overlaying the mixed layer (Fig. 2a). As the atmospheric temperature increases, the ocean's upper layer warms, melting increases and the freshwater surface layer becomes increasingly thicker (Fig. 2b). As time progresses, vertical diffusion further erodes the winter water
layer, and the low-salinity upper layer may reach 50 to 60 m depth at the end of summer (Fig. 2c). Thus, the salinity deficit between the surface and the top of the Winter Water layer is a clear signature of recent freshwater input, from which freshwater thickness can be derived.





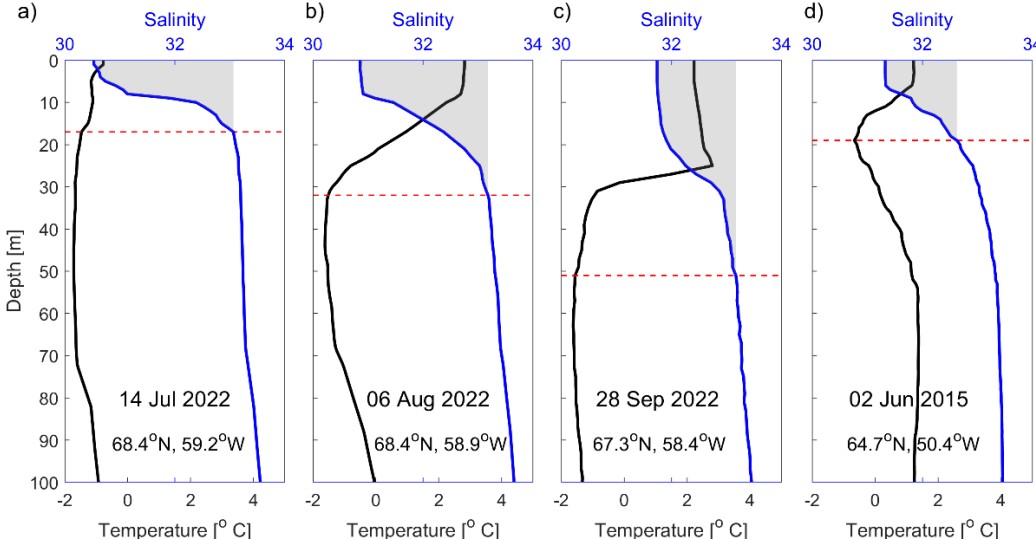

**Fig. 2.** Evolution of the freshwater surface layer throughout summer in Baffin Bay. Profiles of salinity (blue line) and temperature
(black line) in (a) early, (b) mid, and (c) late summer. The thin dashed red line represents the top of the winter water layer, which is
deepening as the season progresses. The gray shade represents the salinity deficit *Δ*$_S$ from which the summer freshwater thickness
is estimated following Eq. (1). In (d), the profile illustrates a case where the top of the winter water layer is characterized by a
temperature minimum.

Using a mass balance formula, and considering that the main source of summer freshwater is ice, the summer freshwater
thickness (SFT) is estimated as follows:

$$SFT = \frac{\rho_w}{\rho_i} \cdot \Delta_S / (S_D - \Delta_S - S_i) \quad \text{(Eq. 1)}$$

with $\Delta_S = \int_0^D \delta S(z) - S_W dz$, and where, $\rho_w$ =1025 kg m$^{-3}$ is the density of seawater and $\rho_i$ that of sea ice. $S_i$ is the sea ice
salinity, $D$ is the depth of the base of the salinity deficit layer (or the top of the winter water layer), $S_w$ is the salinity of the
winter water, and S(z) is the salinity at each depth. We use two different sets of values for $\rho_i$ and $S_i$, for the offshore and near-
shore regions. In the offshore regions of Baffin Bay, we assume that most of the freshwater input originates from sea ice, and
therefore we adopt standard values for sea ice in the Northern Hemisphere with $\rho_i$ =910 kg m$^{-3}$ and $S_i$ =4.6. In near-shore
regions, we assume that the freshwater released from river and glacier runoff flows along, and stays confined near the coast
because the flow in polar regions tends to follow the isobaths. We infer, from the results of the freshwater budget (section 3.2),
that the freshwater detected by our method in the near-shore regions is a mix of approximately 1/3$^{rd}$ melted sea ice and 2/3$^{rd}$
melted glacier ice and river water. Accordingly, we set $\rho_i$ = 977 kg m$^{-3}$ and $S_i$ = 1.53 for all data acquired where the bottom
depth is shallower than 500 m, i.e., the limit of the shelf break (Fig. 1).

## 2.1.3 Implementation

Implementing Eq. (1) requires that only the temperature and salinity profiles with a clear freshwater input signature are
selected. The following criteria are used to select these profiles:





(1)   The temperature minimum in the upper 100 m should be less than 0°C. This criterion ensures that winter water still exists in the water column (Fig. 2).

     (2)   A temperature gradient of at least -0.1°C m$^{-1}$ exists within the profile. This ensures that the surface temperature is relatively warmer than the underlying layer, indicating that most ice in the vicinity has already melted (Fig. 2).

     (3)   A difference of at least 0.1 exists between the maximum and minimum salinity in the upper 50 m of the profile. This

165          ensures that freshwater input exists. This condition allows us to exclude the profiles acquired near the beginning of the melting season where little freshwater has been released.

     (4)   No significant positive temperature gradient (increase in temperature with depth > 0.075°C m$^{-1}$) exists between the surface and the top of the winter water layer. Such gradient is considered a signature of a warm and salty water mass subsurface intrusion, which would impair our estimate.

(5)   Profiles acquired in the vicinity of high (>25%) sea ice concentration are discarded, as such profiles are considered to have been acquired before the end of the ice melting.

        Lastly, the top layer of the winter water, or the base of the salinity deficit layer $D$ (Eq. 1), is selected by using a temperature gradient criterion. For profiles without a clear temperature minimum and a minimum temperature less than -1°C, $D$ is the depth below which a temperature gradient smaller than -0.04°C m$^{-1}$ is found continuously for 20 m (Fig. 2b, 2c). For

profiles with a clear temperature minimum (Fig. 2d), $D$ is the depth of the temperature minimum. Following the application of these criteria, 11656 profiles out of the 47942 original profiles were selected and yielded freshwater thickness estimates (Fig. 3a). Almost none of the profiles in the Labrador Basin are selected for this study since winter water does not exist in this area (selection Criterion 1). Thus, about 1/3$^{rd}$ of the 34184 profiles located outside the Labrador Basin could be analyzed.

        Of the 11656 analyzed profiles, 3353 are low vertical-resolution bottle profiles with generally less than 10 points

within the upper 100 m. 502 profiles were acquired by Seaglider with about one point every 5 m, and 2097 profiles were acquired via biologging (generally between 10 and 18 points within the upper 100 m). The other 5702 profiles were acquired by high-resolution CTD. The discrepancy between the data sources, with half of our dataset made of low-resolution profiles (defined here as less than 1 data point per 3 m, Fig. 3b), could introduce a bias in our estimate. To estimate such bias, we conducted a test whereby the resolution of all the high-resolution profiles was randomly downgraded to that of any of the low-

resolution profiles. The summer freshwater thickness was then recalculated from these downgraded profiles and compared with that of the original high-resolution profiles. The results revealed that low-resolution profiles underestimate the freshwater thickness by 11%. We estimate that the freshwater thickness and volume estimates presented in section 3 could be under-evaluated by about 5% since half of our dataset comprises these low-resolution profiles. In section 4, the climatological difference in freshwater volume is evaluated for the period before and after 1996. In these two periods, respectively 67% and

39% of the data are low-resolution profiles, which suggests that the results of the period before 1996 might be underestimated by about 3% compared to those after 1996. This small bias does not change the conclusions of this paper.



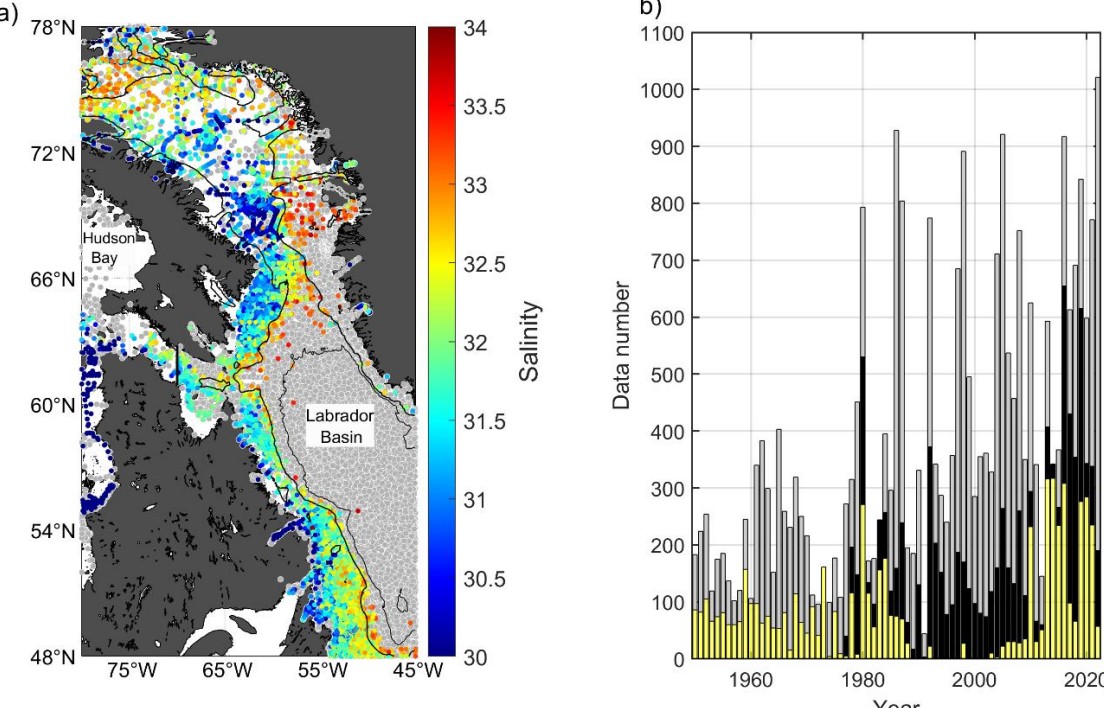

**Fig. 3.** **(a) Distribution of summer salinity data at 10 m depth in Baffin Bay and the Labrador Sea. The colored (grey) dots**
**represent data selected (not selected) for the analysis. The two black contours represent the 500 m and 2500 m isobaths. (b) Data**
**number as a function of the year. The gray bars represent the total number of data (except in the Labrador Basin and Hudson Bay)**
**for a given year, and the black bars represent the number of data selected for analysis in this study. The yellow bars represent the**
**number of analyzed low vertical-resolution (less than 1 point per 3 m) data. Years with only a yellow bar but no black bar indicate**
**years when all analyzed data are low-resolution profiles.**

**2.2 Freshwater budget**

This section describes the methodology used to estimate all the summer freshwater budget components except sea ice volume
and exported sea ice from the Arctic, which are addressed in the next two sections.

**2.2.1 Model and reanalysis data**

The main sources of data for the budget originate from the glacier energy-mass balance model (GLIMB, Fujita and Ageta,
2000; Fujita and Sakai, 2014) for estimating the ice melt volume in ice-covered areas, and an energy balance model developed
by Fujita and Sakai (2014) for the river runoff in non-ice covered areas. Both discharge models are forced by ERA5 daily data
(0.25° resolution, averaged from hourly data), and the air temperature was estimated from pressure level atmospheric
temperature and geopotential height (Khalzan et al., 2022). For the glacier energy-mass balance model, parameter calibrating
precipitation amount (Sakai et al., 2015; Sakai and Fujita, 2017) was estimated at 0.5° resolution to yield the observed long-
term mass balance (20 years between 2000 and 2019) derived from satellite data (Hugonnet et al., 2021). Then the model



calculated the energy and mass balance of the ice surface at a 50-m elevation interval. For the ice-free terrain, the energy balance model was used without precipitation calibration. The rationale underlying the use of calibration for the ice-covered regions and no calibration for the ice-free region is explained in section 2.2.2.

215        To calculate the freshwater input to Baffin Bay, we determined the basins affecting the bay (Fig. 4). Basin boundaries were determined by catchment analysis based on a 500-m resolution digital elevation model (ETOPO2022, 15sec). Glacier mask was used from the Randolph Glacier Inventory version 6 (RGI consortium, 2017). Greenland basins were used from IMBIE (the IMBIE team, 2020) and the glacier and ice sheet masks were merged as ice masks. At each calculation grid (0.25 ° as ERA5 data), hypsometry (area-elevation distribution) was prepared for each ice-covered and ice-free surface at a 50-m elevation interval. For clarity's sake, we summarized the discharge data in Table 1 for the whole study region and the results
for each individual basin are provided in the supplementary materials (Tables S1 and S2).

        The models did not calculate the "ice discharge", which is freshwater input as a form of iceberg mainly from the Greenland ice sheet. Instead, the ice discharge data presented in our study (Table 1) originates from the dataset of Mankoff et al. (2021) with the reported 10% uncertainty (Mankoff et al., 2020). For the ocean surface, monthly precipitation and evaporation data of ERA5 were used (Hersbach et al., 2023). Precipitation minus evaporation (P-E) yields freshwater input
from the atmosphere. All the data described in this section were integrated from April to August in the Labrador Sea and from January to September in Baffin Bay, assuming that the freshwater inputs during these periods remain in the study region until the end of summer. The data were averaged over the period 1965-2022, except for the ice discharge (1986-2022). In both cases, we assume that these averaged periods represent well the situation for the overall 1950-2022 period which the ocean freshwater thickness data covers. In Table 1, the uncertainties in volume for P-E over the ocean are derived from the ERA-5 uncertainties for P-E. For the ice melt and river runoff, the uncertainties also originate for the ERA-5 uncertainties for P-E
integrated over the land regions, since these data are used as input for the models.



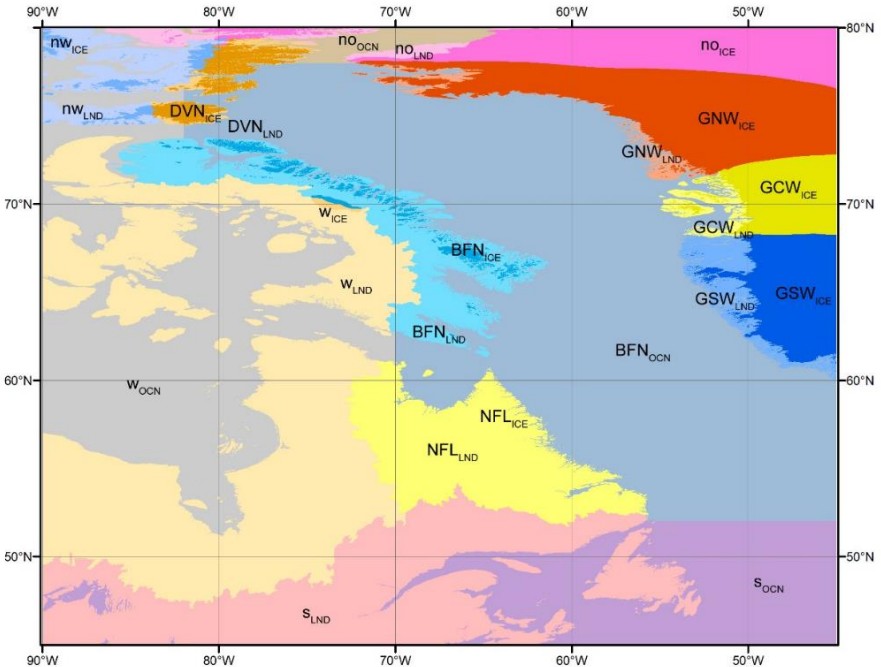

**Fig. 4.** **Domains and basins for calculating freshwater input to Baffin Bay. BFN, NFL, and DVN denote Baffin, Newfoundland, and Devon regions. GNW, GCW, and GSW denote the Northwest, Central West, and Southwest basins of the Greenland ice sheet.**
**Regions expressed with small characters as no, nw, w, and s denote north, northwest, west, and south regions that would not affect the water budgets of Baffin Bay. Subscripts ICE, LND, and OCN denote ice surface, ice-free terrain, and ocean surface.**

### 2.2.2 Calibration of the models

We adopted the simulated freshwater inputs from the ice-covered surface with calibrated precipitation and from the ice-free
surface with non-calibrated precipitation, respectively. To discuss the validity of this procedure, we compared the results
(discharge and precipitation) with vs. without the calibration (Fig. 5).

For the ice-free terrain, we first compared the results yielded from the monthly ERA5-based precipitation minus
evaporation, and those of the simulation without the precipitation calibration (Figs. 5a-c). The precipitation data from the two
sources are identical (Fig. 5b) and the slight differences in discharge or river runoff (Fig. 5a) thus result from the moderate
discrepancy in the evaporation results from the two sources (Fig. 5c). Further, the comparison between the simulated results
in their calibrated and non-calibrated version (Fig. 5d-f) revealed that the calibrated discharge data were two to four times
larger than the non-calibrated results (Fig. 5d). Since the evaporation data exhibit little sensitivity to the calibration (Fig. 5f),
the discharge responds mostly to the precipitation calibration (Fig. 5d vs. Fig. 5e). The largest contribution to the river runoff
is that of the Newfoundland and Labrador region (NFL in Fig. 5), with 301 km$^3$ for the non-calibrated data vs. 1100 km$^3$ for
the calibrated data. Since discharge data estimated from in-situ data by Dery et al. (2016) in this same region represents 190
km$^3$, we decided to adopt the simulated discharge without precipitation calibration as the freshwater input from the ice-free
terrain.





On the other hand, the precipitation calibration did not affect the discharge from the glacier (Fig. 5g). This seems counterintuitive because the GLIMB model should yield more negative mass balance than the satellite-based observational data if the lower ERA5 precipitation was used directly. Thus, the model should yield more meltwater as discharge. Conversely,

as the precipitation calibration resulted in an increase in precipitation, the meltwater discharge should be suppressed. The insignificant difference between the results with/without precipitation calibration is due to the compensation of the suppressed meltwater by the precipitation falling as rainwater during the summer season. We thus decided to use the precipitation-calibrated version of the GLIMB model results for our study.

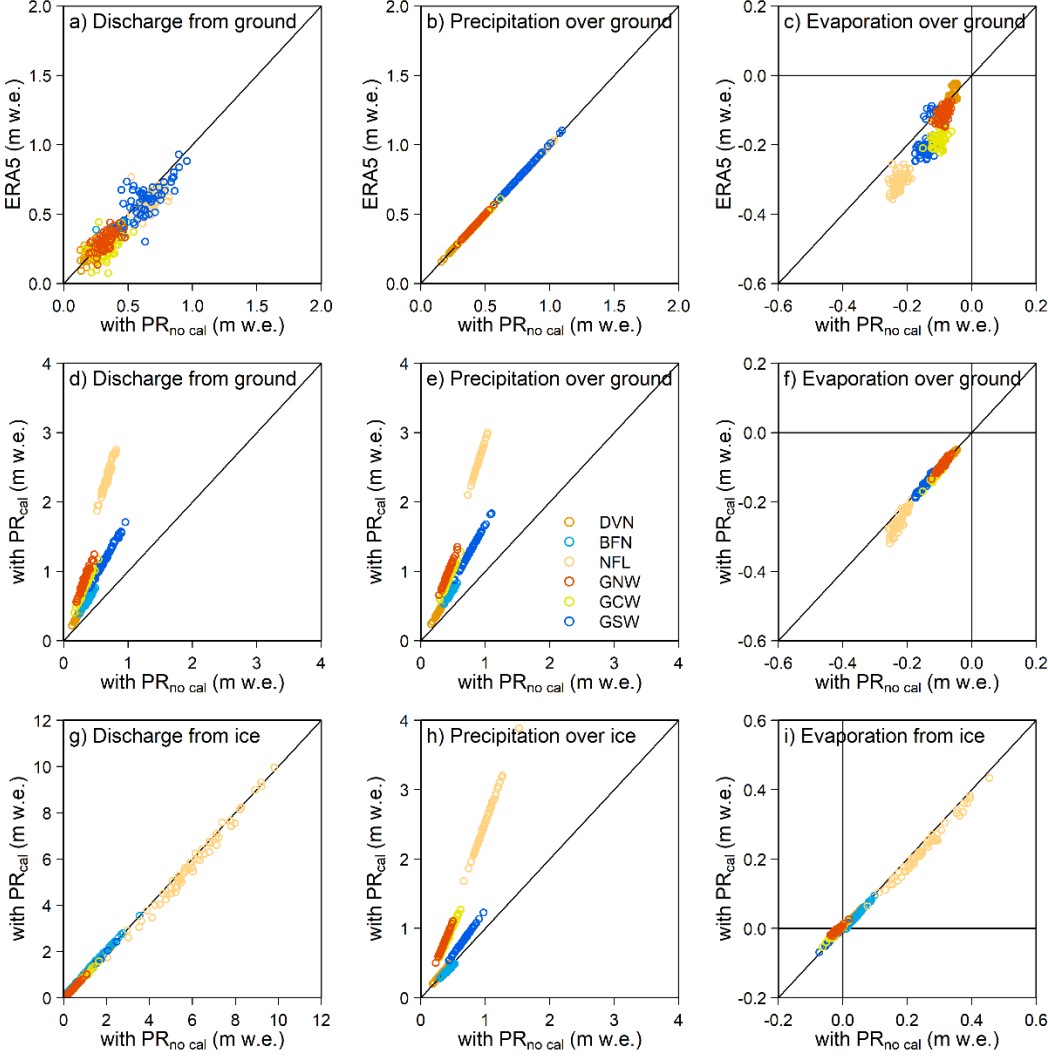

**Fig. 5.  Comparisons of discharge (left panels, a, d, g), precipitation (center panels, b, e, f), and evaporation (right panels, c, f, i) between (a-c) the simulation without precipitation calibration and the use of ERA5 data, (d-f) between the simulations with vs. without precipitation calibration for the ice-free terrain, and (g-i) for the glacier surface. The colors of the dots indicated in (e) correspond to each of the subregions in Fig. 4.**




**2.3 Sea Ice Thickness**

We primarily used the SMOS-CryoSat-2 merged product developed by Ricker et al. (2017), because this product has the advantage of using the daily temporal resolution and higher accuracy for thin ice of the SMOS radiometer, and the high accuracy for thick ice of CryoSat-2's altimeter. We estimated a climatology of March sea ice volume for the period 2011-2022 (Fig. 6a), and the thickness values multiplied by the March sea ice concentration were integrated to get a climatological value of sea ice volume of 749 km$^3$ (Table 1). The uncertainties in sea ice thickness provided in the merged product were integrated

similarly, yielding a volume uncertainty of 154 km$^3$.

To better understand longer-term variations of the sea ice volume in our study region, we used the proxy sea ice thickness product developed by Glissenaar et al. (2023), which provides a monthly record of winter sea ice thickness data from 1996 to 2020. This dataset was obtained via a regression model combining information from the Canadian Ice Service ice charts with scatterometer backscatter data from various satellites, and using machine learning to train the model with CryoSat-

2 data. This product covers all regions of the Canadian Arctic.

The 2012-2020 climatology of proxy sea ice thickness (Fig. 6b) does not compare well qualitatively with the SMOS-Cryosat-2 climatology (Fig. 6a), possibly because the proxy product is trained with CryoSat-2, which has a poor detection capability for thin ice, whereas this shortcoming is compensated by SMOS data in the SMOS-CryoSat-2 merged product. Quantitatively, for similar areas between 63°N and 78°N, the 10-year average March volume is approximately 50% higher for

the proxy product (944 km$^3$) than for the SMOS-CryoSat-2 product (628 km$^3$). However, the time series from the two data sources exhibit similar variations (Fig. 6c). Thus, using the proxy sea-ice product to estimate the variability of sea ice between 1996 and 2020 is a reasonable approach. The time series of the March proxy ice thickness in Fig. 6c and Glissenaar et al. (2023) exhibit a significant decreasing trend. This implies that our climatological value of 749 km$^3$, estimated for the period 2011-2022, may be an underestimate considering that our freshwater thickness climatology spans the period 1950-2022.



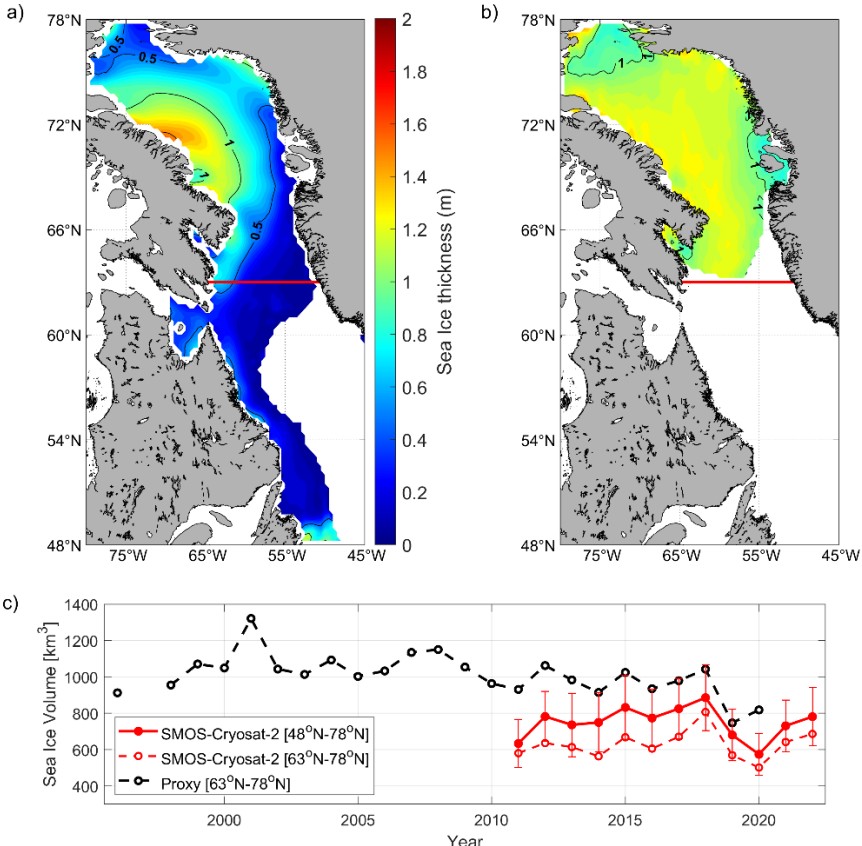

**Fig. 6.** **March sea ice thickness distribution averaged for (a) the SMOS-Cryosat-2 merged product (2011-2022), and (b) the sea ice thickness proxy product (2011-2020) by Glissenaar et al. (2023). (c) Time series of sea ice volume from the SMOS-Cryosat-2 product for the whole study region (solid red line), and from the two products, integrated over the Baffin Bay only (black-dashed and red-dashed curves). The error bars represent the uncertainties in sea ice thickness provided in the SMOS-Cryosat-2 product.**

**2.4 Observed ice volume flux from the Arctic Ocean and Canadian Arctic Archipelago**

**2.4.1 Methodology**

We use a widely established methodology to estimate the volume flux of sea ice transported from the western Arctic Ocean to Baffin Bay and the total volume of ice exiting from Baffin Bay to the Labrador Sea from October 2016 to December 2022 based on previous studies (e.g., Agnew et al., 2008; Kwok et al., 2010; Howell and Brady, 2019; Moore et al., 2021). Briefly, overlapping pairs of satellite imagery from RADARSAT-2, Sentinel-1, and the RADARSAT Constellation Mission (RCM) were acquired across the width (L) of the Nares Strait (139 km), Jones Sound (48 km), Lancaster Sound (83 km), and Davis Strait (450 km) and resampled to 200 m spatial resolution. The first three gates allow us to estimate the volume of ice coming into Baffin Bay, and the last one the volume out of Baffin Bay and into the Labrador Sea. For each SAR image pair, sea ice motion was estimated using the Environment and Climate Change Canada Automated Sea Ice Tracking System (ECCC-





ASITS; Howell et al., 2022). Sea ice motion estimates were interpolated to a 30 km buffer region at each gate and then sampled

at 5 km intervals. The sea ice area flux ($F_A$) was then calculated using the following equation:

$$F_A = \sum c_i u_i \, \Delta x \qquad \text{(Eq. 2)}$$

where, $\Delta x$ is the spacing along each gate (i.e., 5 km), $u_i$ is the ice motion normal to the flux gate at the $i^{th}$ location and $c_i$ is the

sea ice concentration determined from the Canadian Ice Service ice charts (Tivy et al., 2011).

305         The uncertainty ($\sigma_{FA}$) in $F_A$ can be estimated following Kwok and Rothrock (1999) by assuming errors in sea ice

motion are additive, uncorrelated, and normally distributed using the following equation:

$$\sigma_{FA} = \frac{\sigma_u L}{\sqrt{N_s}} \cdot \sqrt{N_D} \qquad \text{(Eq. 3)}$$

where, $\sigma_u$ is the error in SAR-derived ice motion (0.43 km; Komarov and Barber, 2013), $L$ is the width of the gate, and $N_s$ is

the number of samples across the gate, and $N_D$ is the number of observations per month (~30). Accordingly, the monthly $\sigma_{FA}$

estimates for Nares Strait, Davis Strait, Jones Sound, and Lancaster Sound are 62 km$^2$, 112 km$^2$, 40 km$^2$, and 48 km$^2$,

respectively.The volume flux ($F_V$) was obtained by multiplying the area by the monthly CryoSat-2 sea ice thickness from

Landy et al. (2022). The uncertainty ($\sigma_{FV}$) in volume flux can be estaimtes following Kwok and Rothrock (1998) using the

following equation:

$$\sigma_{FV} = \sqrt{(F_A \sigma_h)^2 + (h \sigma_{FA})^2} \qquad \text{(Eq. 4)}$$

where, $h$ is the ice thickness and $\sigma_h$ is the uncertainty in thickness is taken from Landy et al. (2022). Accordingly, the monthly

$\sigma_{FV}$ estimates for Nares Strait, Davis Strait, Jones Sound, and Lancaster Sound are 3 km$^3$, 19 km$^3$, 1 km$^3$, and 2 km$^3$

respectively.

**2.4.2 Sea ice volume flux climatology and monthly mean**

In our study, we use the ice volume fluxes for two purposes: (1) to provide a climatological value of ice volume flux to Baffin

Bay for the freshwater budget, and (2) to estimate the amount of locally-produced volume of ice transported from Baffin Bay

to the Labrador Sea. Regarding (1), we considered that the March sea ice volume (section 2.3) includes both locally-produced

sea ice (e.g. in the North Water Polynya), and ice exported in winter through Nares Strait as well as Jones and Lancaster

Sounds. However, the ice volume fluxes to Baffin Bay occurring from March onward are not accounted for. Thus, we use the

ice volume fluxes data to estimate the additional amount of sea ice transported into Baffin Bay between March and the end of

our sampling period in September. We first integrate the ice volume fluxes at all three gates between March and September

for each year between 2017 and 2021. The climatological value for the freshwater budget is then estimated by averaging the 5

years of estimates. The average March-September sea ice volume flux for the 5 years is 91 km$^3$, about half of the yearly

volume.

        In regard to (2), we assumed that the difference between the ice volume flux out of Baffin Bay (Davis Strait gate) and

that into Baffin Bay (Nares Strait, and the Jones and Lancaster Sounds gates) should correspond approximately to the amount



of sea ice locally produced in Baffin Bay and transported out to the Labrador Sea. We estimate the monthly mean of this difference averaged for the period 2017-2021 (section 4.2).

## 3 Results

### 3.1 Summer freshwater thickness climatology

The meltwater thickness was obtained for each selected profile from Eq. (1) and its distribution for the period 1950-2022 is displayed in Fig. 7a. These data were then mapped using the shrinking/stretching constraint integrated mapping scheme of Mensah and Ohshima (2023). The mapping principle relies on the conservation of (barotropic) potential vorticity as the main constraint for flows in polar and subpolar regions, resulting in the water column flowing along the isobaths. The mapping scheme consists of a weighted mean with two parameters, i.e., the distance $r$ between the grid point and a given observation,

and the shrinking/stretching constraint $\varPhi = \frac{|\varDelta h|}{h}$ where $h$ is the grid point's bottom depth and $\varDelta h$ is the difference between the bottom depth at the grid point and that at the observation location. The decorrelation scales estimated using Mensah and Ohshima's (2023) methodology are $L_r$=157 km for distance and $L_\varPhi$=0.51 for the shrinking/stretching parameter. The 95% confidence interval was estimated for each grid cell as the root mean square difference between the grid cell value and the observed values within one decorrelation scale of each grid cell, similar to Mensah and Ohshima (2023).

345         The mapped distribution of freshwater thickness in Fig. 7b illustrates how the influence of sea ice melt, glacier melt, river runoff, and exported Arctic sea ice combine and affect the distribution. At the basin scale, the distribution of summer freshwater thickness is qualitatively consistent with that of the maximum sea ice thickness from the SMOS-Cryosat-2 product (Fig. 6a). The two distributions exhibit relatively low values north of 74°N, which is likely due to the presence of the North Water polynya, and maximum values east of Baffin Island owing to the accumulation of sea ice in this region (Landy et al.

2017). The lowest freshwater thickness values are found east of Central Greenland and south of 62°N along the coast of Newfoundland. The overall similarity between the freshwater thickness and the sea ice thickness distributions suggests that local sea ice production/melt is a dominant factor in the summer freshwater budget in this region.

         However, smaller-scale features and quantitative differences highlight the role of other freshwater sources. For example, the highest freshwater thickness values (>3 m) are found along the coast of southern Newfoundland between 53°N

and 56°N, just outside of the Churchill River estuary (see Fig. 7b for the location). The Churchill River is the largest river in our study region and its runoff is 57 km$^3$ yr$^{-1}$ (Déry et al., 2016). Integrating the freshwater thickness greater than 2 m between 53°N and 56°N yields a volume of 54 km$^3$. Similarly, the location of high (> 2m) freshwater thickness off the fjords of Baffin Island (between 69°N and 72°N) is consistent with that of the main glaciers foot on the island. Higher freshwater thickness values off the coast of southwestern (between 63°N and 66°N) and northwestern Greenland (north of 75°N) are also consistent

with the locations of large Greenland glaciers (Mankoff et al., 2020), such as Narsap Sermia in the southwest, or Paakitsup Sermersua (Pituffik Gletsjer) and Savissuup Sermia (Savigssuaq Gletscher) in the northwest (see Fig. 7b for the locations).



The freshwater thickness climatology (Fig. 7b) provided good qualitative results, that enable us to distinguish several freshwater sources. However, a comparison between the freshwater volume integrated from our climatology and the freshwater budget obtained from the sources listed in Section 2 is now necessary for a quantitative assessment.

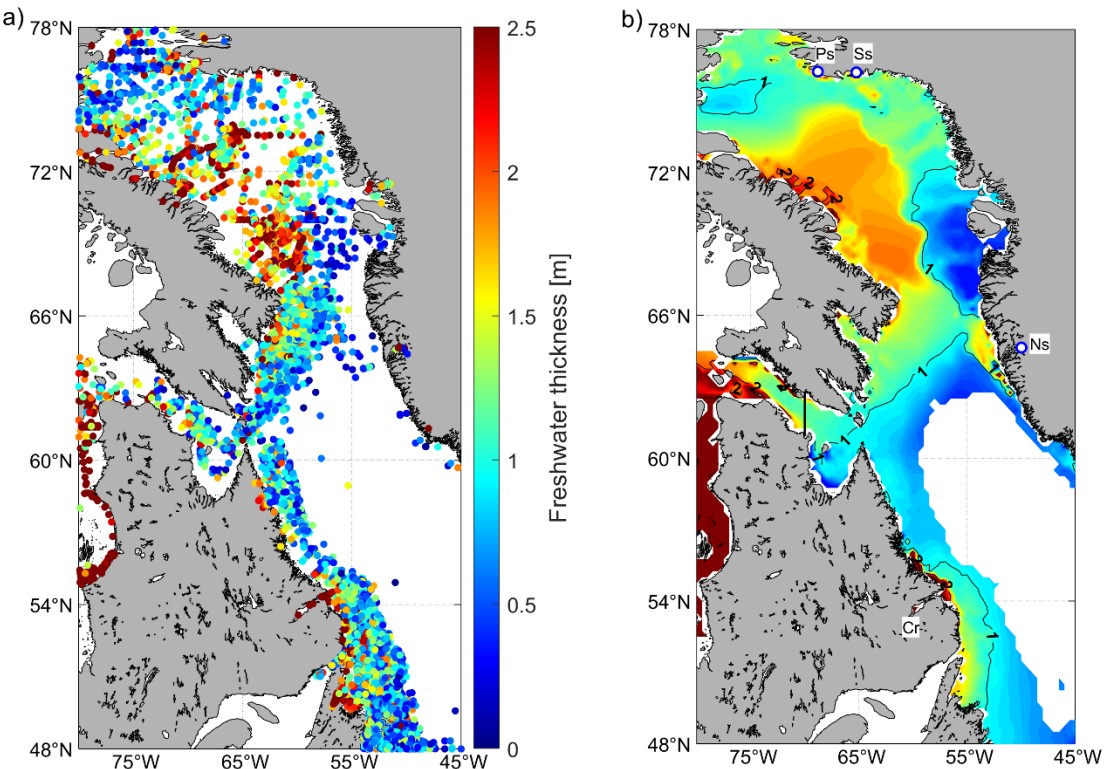


**Fig. 7.** **(a) Distribution and (b) climatology of summer freshwater thickness for the period 1950-2022. The data in the region west of the black line in the Hudson Strait is not counted in the freshwater volume estimate. The initials on (b) indicate the Churchill River (Cr), and a few of the western Greenland Glaciers, Narssaap Sermia (Ns), Savissuup Sermia (Ss), and Paakitsup Sermersua (Ps). The names and locations of the glaciers were taken from Black and Joughin (2023).**

## 3.2 Summer freshwater budget

The integration of the summer freshwater thickness climatology yields a total volume of 1956 ($\pm$303) km$^3$. The contributions of freshwater input by locally produced sea ice, exported sea ice from the Arctic, P-E over the ocean and on land, glacier melting, and iceberg calving are listed in Table 1 and sum up to a total summer freshwater budget of 2286 km$^3$. The good comparison with the freshwater volume climatology suggests that our methodology is well adapted both qualitatively and

quantitatively to estimate summer freshwater input. The largest contribution to the summer freshwater budget is locally produced sea ice with 749 km$^3$ or 33% of the total. If sea ice exported from the Arctic Ocean into Baffin Bay (91 km$^3$) is added, the total contribution of sea ice to the freshwater budget is 37%. The next largest contribution is the ice from the glaciers by melting and iceberg calving (602 km$^3$, 26%). The contribution of river runoff or P-E on land (458 km$^3$, 20%) and over the ocean (386 km$^3$, 17%) are practically equal. These results illustrate how the contribution of land- or sea ice is crucial in the



freshwater budget of the region. They represent 63% of the total summer freshwater budget and are more likely to be significantly affected by long-term climate change compared to P-E.

A similar estimate for the period 1996-2022 (during which all data for each of the budget contributions are available) yields 1990 km$^3$ for the climatology of SFV from hydrographic data and 2394 km$^3$ for the freshwater budget from each component. Thus, regardless of the period, the summer freshwater volume is slightly smaller than the freshwater budget.

Several factors may explain or contribute to this discrepancy: (1) the ~5% underestimate of freshwater volume due to the use of low-resolution profiles (section 2.1), (2) an overestimate of some of the freshwater budget terms, and (3) the sparse ocean data used for the climatology. Among the freshwater budget terms, the estimation of P-E from reanalysis data has a large uncertainty as pointed out by Haine et al. (2015). We also note that the runoff for Labrador and Newfoundland from our model is 301 km$^3$ (Fig. 1) whereas direct estimates of river runoffs for this region yield 190 km$^3$ (Dery et al., 2016). Last, the sparse

nature of the ocean data used for the climatology could also be a cause for misestimates. For example, there is no data available for some of the Fjords off central Baffin Island (69°N) where large glaciers are terminating.

To avoid adding uncertainty, the western part of the Hudson Strait (Fig. 1), where large freshwater values are found along the northern coast of Labrador (Fig. 7b) is excluded from the analysis. We indeed considered that this area is influenced by the freshwater outflow from Hudson Bay, which is not straightforward to quantify. Our data suggest that the freshwater

plume from Hudson Bay has not reached beyond the Hudson Strait by summer's end.





**Table 1: Summer freshwater budget for local freshwater sources in Baffin Bay and the Labrador Sea. Uncertainties are shown between brackets.**

| Components | Freshwater volume (km$^3$) | % | Freshwater volume 1965-1995 (km$^3$) | Freshwater volume 1996-2022 (km$^3$) | Volume Difference After - Before 1996 |
|---|---|---|---|---|---|
| Sea Ice | 749 (±154)[1] | 33 | - | 749 (±154) | **-** |
| Exported Arctic Sea Ice | 91 (±6)[2] | 4 | - | 91 (±6) | **-** |
| P-E Ocean | 386 (±26) | 17 | 389 (±26) | 383 (±25) | **-6** |
| River runoff (P-E Land) | 458 (±51)[3] | 20 | 453 (±58) | 463 (±45) | **+10** |
| Glacier melt | 428 (±48) | 19 | 348 (±44) | 519 (±50) | **+171** |
| Ice discharge (Iceberg calving) | 174 (±17)[4] | 7 | 133 (±13)[5] | 189 (±19) | **+56** |
| Total | **2286 (±302)** | **100** | 2163 (±301) | 2394 (±299) | **+231** |


Notes: 1 Data available from 2011 to 2022, 2 Data available from 2017 to 2021, 3 Out of 458 km3, 301 km3 are contributed by Newfoundland and Labrador, excluding the St-Lawrence River.

4 Data from Mankoff et al. (2021), from 1986 to 2022. 5 Average from 1986 to 1995.

## 4 LONG-TERM CHANGES IN SUMMER FRESHWATER INPUT

To evaluate the effects of multi-decadal changes in freshwater thickness in the Baffin Bay and the Labrador Sea, two climatologies for the period before and after 1996 were estimated from the freshwater thickness data (Fig. 8a-b). The year

1996 was chosen because a decreasing trend in the winter sea ice extent of the Baffin Bay/Labrador Sea area starts from that year (Cavalieri and Parkinson, 2012). The difference between the two climatologies (Fig. 8c) highlights the magnitude and spatial variability of these changes. A contrast exists between Baffin Bay, where the summer freshwater thickness exhibits a clear increase (+226 km$^3$) except in the area around 70°N, and the Labrador Sea, where a significant decline (-112 km$^3$) is seen. In the next two sections, the changes in Baffin Bay and the Labrador Sea will be analyzed separately and compared with

changes in the local sources' freshwater budget. In the following analysis, the Baffin Bay and the Labrador Sea are defined as the areas located north and south of 62°N, respectively.



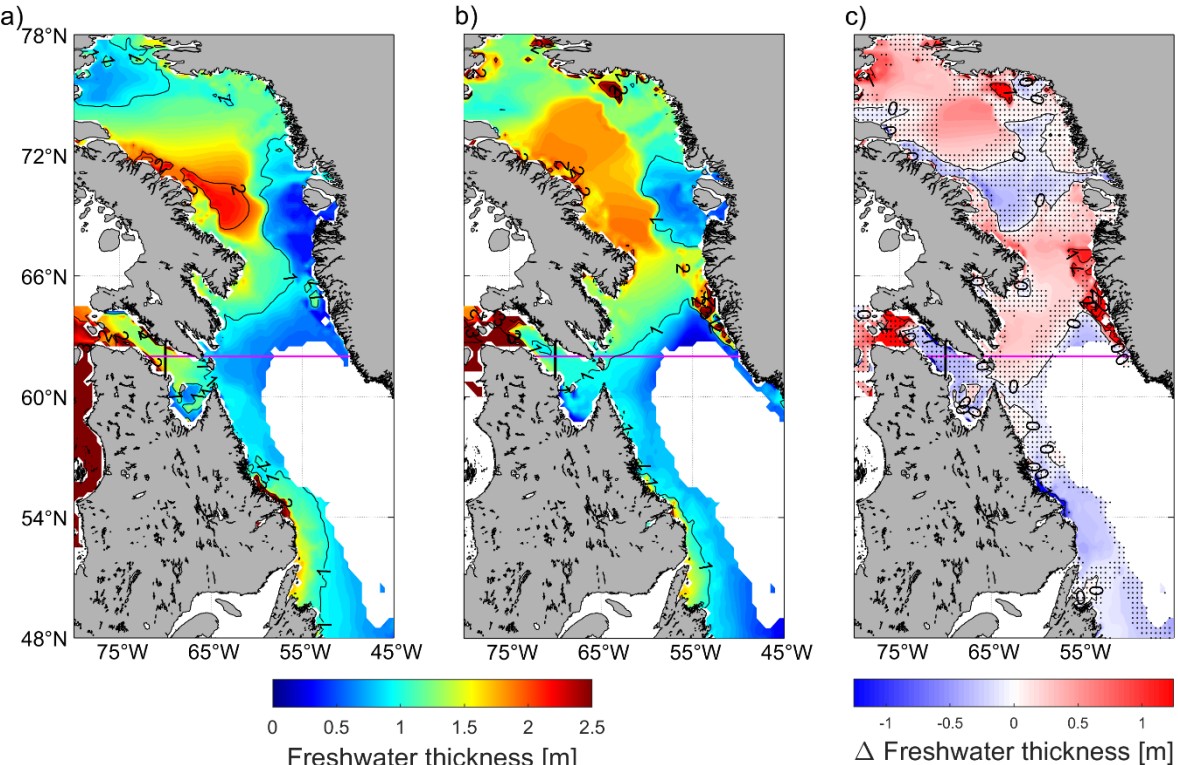

**Fig. 8.** **Climatology of summer freshwater thickness for the periods (a) 1950-1995 and (b) 1996-2022. (c) Climatological difference between the two periods (b minus a). The dotted areas represent the location where the difference is below the 95% confidence level estimated following Mensah and Ohshima (2023). The pink line along 62°N represents our defined limit separating Baffin Bay to the north and the Labrador Sea to the south.**

## 4.1 Baffin Bay

The climatological difference between the periods after and before 1996 exhibits positive values, except in the region between 70°N and 72°N, where a significant decrease in summer freshwater thickness exists. This area is located south of the North Water polynya, and the decline in freshwater thickness in this region could be the signature of the decreasing sea ice volume trend in Baffin Bay (Glissenaar et al., 2023, and section 4.2).

The volume difference between the two climatological periods north of 62°N is an increase of 226 km$^3$. The difference in the freshwater budget in the whole area before and after 1996 except for sea ice thickness and export of Arctic sea ice (Table 1) yields 231 km$^3$. This difference is almost entirely due to the increase in glaciers melting and iceberg calving (+227 km$^3$, Table 1), most of these glaciers being located north of 62°N. An increase in the ice volume flux through Nares Strait should also affect the region, although a rigorous estimate cannot be established. The duration of the ice arches which prevent ice from being transported south of the Nares Strait has decreased notably after 2007, and Moore et al. (2021) results revealed an increase of 70% of the ice volume flux between 1997-2009 and 2017-2019. Presumably, the ice volume flux was significantly lower before 1996 and may have risen by about 40 km$^3$ (considering the 70% increase) between the two climatological periods.



This increased flux could contribute to the larger freshwater thickness north of 74°N in Baffin Bay (Fig. 8c) but is still an order of magnitude below the extra volume of freshwater supplied by glaciers. Thus our results (Table 1) demonstrate that most of the additional summer freshwater volume in Baffin Bay is due to the increase in Greenland, Devon Island, and Baffin Island glaciers melting or iceberg calving, which is consistent with our freshwater volume difference estimate (Fig. 8c).

## 4.2 Labrador Sea

The total volume of summer freshwater south of 62°N decreased by 16% from 716 km$^3$ before 1995 to 604 km$^3$ after 1996. The time series of freshwater thickness anomalies from 1950 to 2022 (Fig. 9b) reveals a moderate decreasing trend of 20 km$^3$ per decade since 1950, with a stronger decline in the second half of the time series. The only freshwater contribution whose volume may have decreased after 1996 is that of sea ice (Table 1, Fig. 6). The sea ice volume present in the Labrador Sea at the end of winter represents only 16% of the total volume in both Baffin Bay and the Labrador Sea (Fig. 6a), thus we will

consider the sea ice volume in the whole study region. Glissenaar et al. reported a strong decreasing trend (1.46 cm/year) in Baffin Bay winter sea ice thickness between 1996 and 2020. We thus compare the time series of freshwater volume in the Labrador Sea after 1996 with that of March sea ice volume derived from the proxy sea ice thickness product (section 2.3). The two time series (Fig. 9c) are only moderately correlated (0.33) but exhibit remarkable similarities between 2012 and 2020 and have comparable decreasing trends, with 54 km$^3$ per decade for the sea ice volume and 72 km$^3$ per decade for the freshwater

volume.

The general agreement between the two time series implies that the sea ice volume in the whole study region plays a major role in influencing the summer freshwater volume in the Labrador Sea. For this to be true, most of the sea ice present in Baffin Bay in March should have been transported and melted to the Labrador Sea by the end of the summer. To verify this possibility, we calculated the 2017-2021 monthly mean of the ice volume flux in (Nares Strait, the Jones and Lancaster Sounds)

and out (Davis Strait) of Baffin Bay (Fig. 10). The volume flux out of Baffin Bay (blue line) is larger than that into the bay (red line). The large difference suggests that besides the ice volume entering Baffin Bay every month, large volumes of ice produced in Baffin Bay also exit through the Davis Strait. The monthly volume of locally produced ice exiting Baffin Bay is inferred from the difference between the volume in and that out of Baffin Bay (green shading in Fig. 10). Between March and July, the cumulated difference is 477 km$^3$, which is in good agreement with the 544 km$^3$ volume of ice present in March north

of the Davis Strait Gate, as estimated from the SMOS-Cryosat-2 data during 2017-2021. We thus conclude that most of the sea ice present in Baffin Bay at the end of winter is transported to the Labrador Sea throughout the spring and summer, and significantly affects the summer freshwater volume variability in this sea. Note that the yearly cumulated ice volume difference (Fig. 10) equals 775 km$^3$, on the higher end of the 500-800 km$^3$ estimated by Kwok (2007) for the period 2002-2007. However, this recent period (2017 to 2021) has been associated with anomalous ice arch collapses in Nares Strait (Moore et al., 2021)

which may contribute to higher volume amounts.





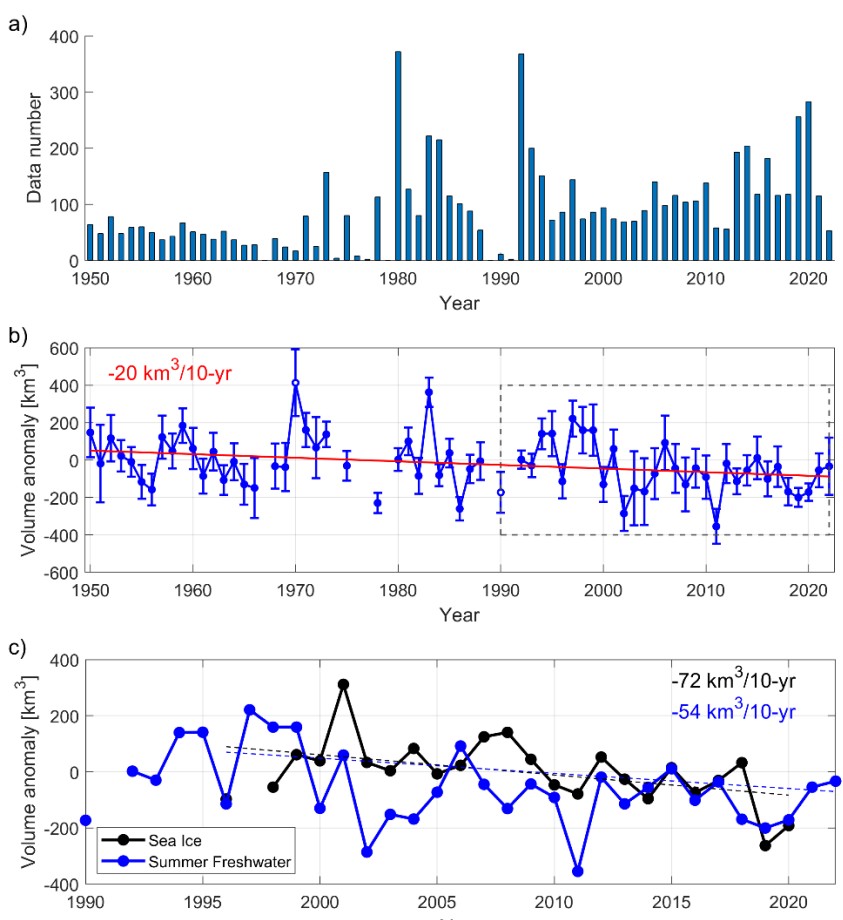

**Fig. 9.** **(a) Number of analyzed summer data in the Labrador Sea south of 62°N for every year between 1950 and 2022. (b) Time series of the summer freshwater volume (SFV) in the Labrador Sea and its trend. The error bars represent the 90% confidence interval estimated following Mensah and Ohshima (2021). (c) Close-up view on the 1990-2022 period of the SFV time series (blue line), plotted together with the March sea ice volume in Baffin Bay. The two trends for the period 1998-2022 are shown in dotted black and blue lines.**

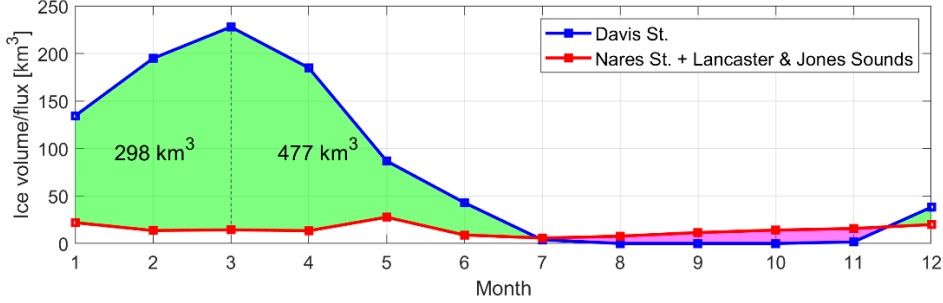

**Fig. 10.** **Monthly mean of the volume fluxes through the Davis Strait (solid blue line) and the sum of the volume fluxes in the Nares Strait and the Jones and Lancaster Sounds (solid red line). The positive difference between the two lines (green shading) corresponds approximately to the southward volume flux of sea ice produced in Baffin Bay north of the Davis Strait gate (67.4°N).**





Last, to determine whether other sources of freshwater (e.g., glacier melt from Baffin Bay) play a significant role in the trend of freshwater volume in the Labrador Sea, we calculated time series of the sum of the anomalies of sea ice volume, or/and glacier melt and calving, or/and P-E. We then estimated the correlation between these time series and the time series of freshwater volume anomaly. The correlation does not increase significantly for any of these combined time series. We suggest

that the non-sea ice freshwater sources in Baffin Bay cannot influence the Labrador Sea summer freshwater budget because they have not reached the Labrador Sea by August. This hypothesis is plausible because that nearly 2/3$^{rd}$ of the glacier melt in Baffin Bay occurs between July and September.

## 5. DISCUSSION

The summer freshwater input into both Baffin Bay and the Labrador Sea has been undergoing large changes on the multi-
decadal scale (Section 4). However, the relative importance of these freshwater inputs on the total freshwater content has yet to be evaluated. The freshwater exported from the Beaufort Sea to Baffin Bay and the Labrador Sea (e.g., Haine et al., 2015; Proshutinsky et al., 2019) is likely to be well mixed within the upper ocean following several months and thousands of kilometers of transport between the two regions. Thus, this freshwater source should exhibit no clear surface signal and cannot be detected by our method. To estimate the relative importance of the summer freshwater volume (SFV, estimated in this
study) vs. the total freshwater content (FWC), we estimate the latter quantity using the "practical approach" following Carmack et al. (2008):

$$FWC = \int_0^{D_{ref}} \left(\frac{S_{ref} - S(z)}{S_{ref}}\right) dz \qquad \text{(Eq. 5)}$$

where, $S_{ref}$ =34.8 is a reference salinity and $D_{ref}$ is the depth where the reference salinity is found. In practice, $D_{ref}$ is constrained to 200 m in our study area because the freshwater from the Beaufort Sea passes through straits in the Canadian Archipelago, whose maximum sill depth is about 200 m. The value of $S_{ref}$ was determined by Carmack et al. (2008) as a value representative
of the upper ocean (50-300 m) salinity below which stratification control is dominated by salinity. The FWC was calculated only for the profiles selected earlier for the summer freshwater thickness estimation (Fig. 3b), rather than for the whole dataset. FWC was then mapped for the two same climatological periods as previously (Fig. 11a-b).





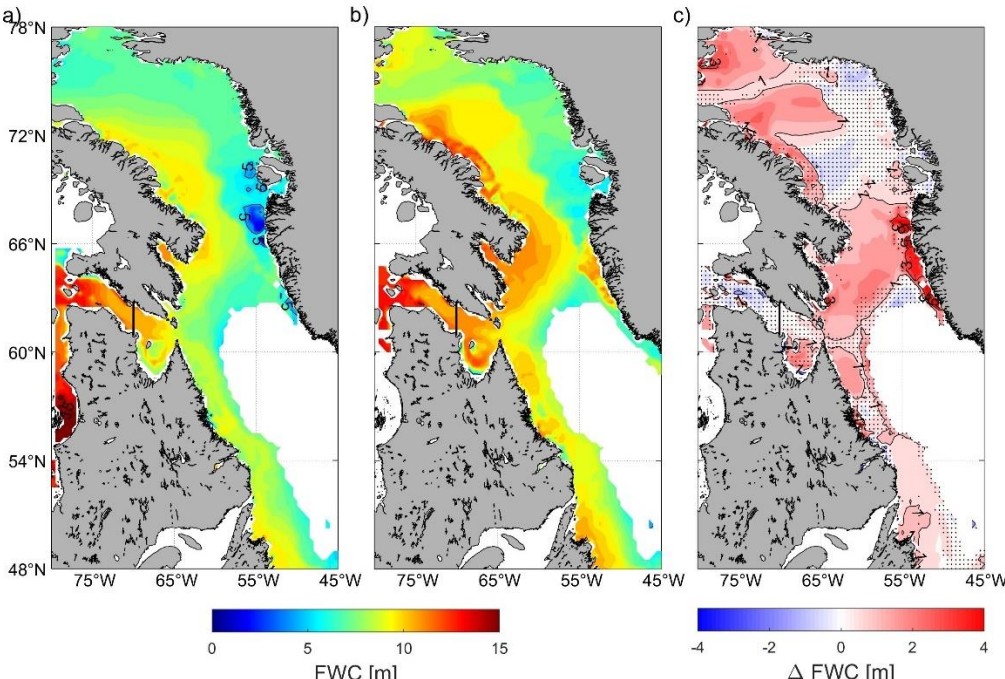

**Fig. 11.** **Climatology of Freshwater Content (FWC) for the periods (a) 1950-1995 and (b) 1996-2022. (c) Climatological difference between the two periods (b minus a). The dotted areas represent the location where the difference is below the 95% confidence level estimated following Mensah and Ohshima (2023).**

The climatological FWC difference map (Fig. 11c) reveals mostly gains in freshwater content. In Baffin Bay, the FWC volume difference integrated north of 62°N represents 1221 km$^3$, against 226 km$^3$ for the difference in SFV, implying that changes in the summer inputs of freshwater may not dominate the variations in total freshwater content at the multidecadal scale. The contrast between FWC and SFV is even more obvious in the Labrador Sea as the trend in freshwater thickness (Fig. 8c, Fig. 9) is negative whereas that in FWC is positive (Fig. 12). The Labrador Sea FWC trend from 1950 to 2022 is +160 km$^3$ per decade against -20 km$^3$ per decade for SFV. A comparison of the 5-year moving-averaged time series (Fig. 12) of SFV and FWC allows us to distinguish two separate phases in the time series. In the first one, from 1950 to 1996, the two time series are highly correlated (0.62, and 0.79 for detrended data), suggesting that SFV is the dominant factor explaining FWC variations at the interannual scale. From 1997 to 2022, the correlation drops to 0.49 (0.54 for detrended data), and the two time series show little similarity between 1998 and 2014.

The opposite sign trends of SFV and FWC, and the poorer correlation between the two time series after 1997 imply that a remote source of freshwater plays a major role in influencing the freshwater content in the Labrador Sea at the multidecadal scale, and episodically at the interannual scale since the end of the 1990s. We argue that the long-term rise of nearly 1000 km$^3$ between 1950 and 2022 (Fig. 12) is likely due to the increase in the Beaufort Gyre freshwater content reported by Proshutinsky et al. (2019). The latter study indicated a growth of Beaufort Gyre FWC of 40% between the climatology of the 1970s and that of the 2003-2018 period. The release of FWC from the Beaufort Gyre is not necessarily linear since it is





mainly governed by the strength of the anticyclonic system over the Beaufort Sea (Proshutinsky et al., 2019). The time series

of Labrador Sea FWC (Fig. 12) indicates a strong increase from 1989 to 1999, which is in reasonable agreement with the

period of "fast release" of FWC determined numerically by Zhang et al. (2021), and lasting from 1983 to 1996.

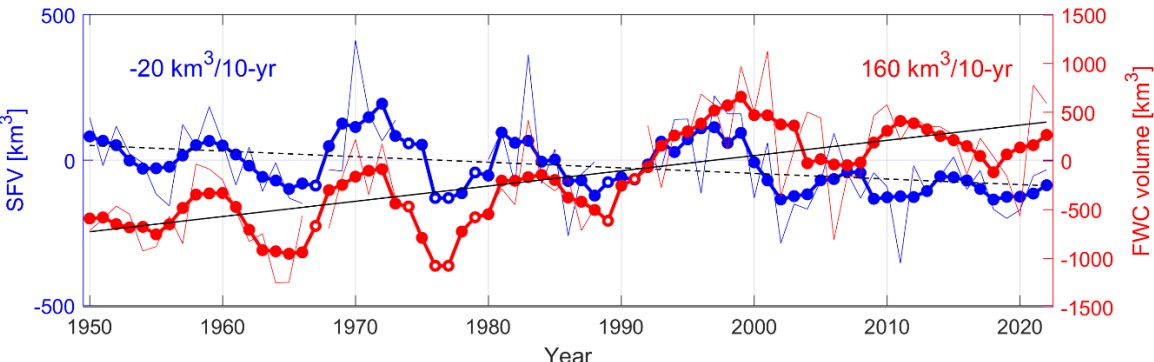

**Fig. 12.    Time series of Summer Freshwater Volume (SFV, blue) and Freshwater Content (FWC, red) anomalies in the Labrador Sea from 1950 to 2022. The thin solid lines represent the raw yearly anomalies and the thick solid lines with dots represent 5-year**
**moving-averaged data. Empty dots represent years without original data. The trends for the FWC and SFV are represented respectively by the thin solid and thin-dashed black lines.**

To further illustrate the effects of the long-term changes in both FWC and SFV on the hydrography of the Labrador

Sea, the salinity profiles averaged from all the data used in this study for the period before and after 1996 are plotted in Fig.

13. In Fig. 13a, the profiles of difference between the salinity at 100 m depth and that at the surface can be considered as a

proxy for the average of all the surface salinity deficit (section 2.1.2, Fig. 2). The profile after 1995 exhibits a notably smaller

salinity deficit, which exemplifies the decrease in summer freshwater input in the Labrador Sea (Fig. 8c, Fig. 9). However, the

0-300 m salinity profiles (Fig. 13b) show considerably more drastic changes. Overall, the whole profile after 1995 shifted

towards lower salinities, with the largest freshening observed between 50 m and 200 m. This freshening corresponds to the

increase in FWC observed in Fig. 11 and in the 1950-2022 trend of Fig. 12. Only the top 30 m of the profile remains unchanged

(Fig. 13b). This is because, near the surface, the freshening is canceled by the decrease in local freshwater input (Fig. 13a).

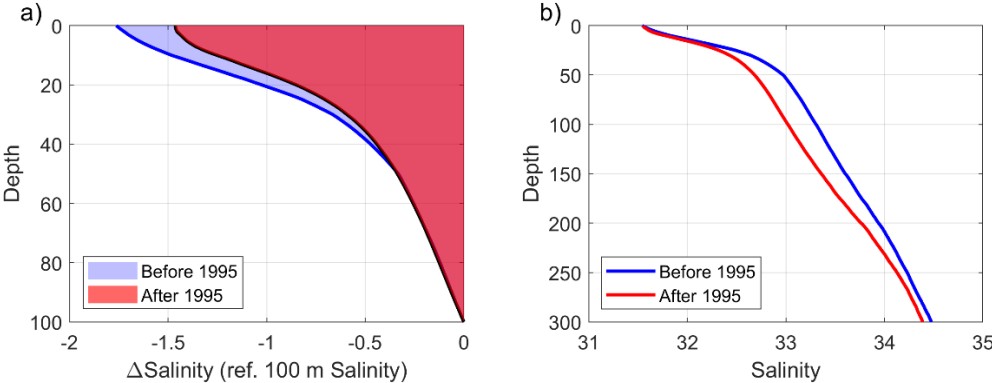

**Fig. 13.    Profiles of (a) salinity difference between salinity and the salinity at 100 m and (b) upper-ocean salinity. The profiles were averaged from the summer data analyzed in this study in the Labrador Sea (south of 62°N) and for the period before and after 1995.**





**In (a), the blue and red shaded areas can be approximated as the surface salinity deficit defined in section 2.2 for the period before**
**and after 1995, respectively.**

Regarding the cause for the discrepancy between the interannual variability in Labrador Sea SFV and FWC after the mid-1990s (Fig. 12), several possibilities exist: (1) a delayed signal from Baffin Bay, (2) a delayed signal from Hudson Bay, and (3) interannual variability of the Beaufort Gyre freshwater release. Our results suggest that besides sea ice, the summer freshwater contributions in Baffin Bay may not reach the Labrador Sea within one season (Section 4.2). However, both the
SFV and the FWC of Baffin Bay could reach the Labrador Sea within one year, and thus variations affecting Baffin Bay could generate an anomaly with a year delay in the Labrador Sea. In fact, during the period 1950-1997, the 5-year moving averaged (detrended) time series of FWC in Baffin Bay and the Labrador Sea are reasonably well correlated at both 0-lag (0.55) and -1-year lag (0.50, not shown). However, after 1997, the 0-lag and -1-year-lag correlation between the two curves is 0.30 and 0.20, respectively. Thus, the delayed FWC signal from Baffin Bay cannot explain the changes in Labrador Sea FWC variability.
This leaves remote sources of freshwater in Hudson Bay (e.g., Granskog et al., 2009) and the Beaufort Sea as the possible governing factors explaining these interannual variabilities. The relationship between interannual variations of the FWC in the Labrador Sea and the Hudson Bay and Beaufort Gyre is outside the scope of this study but merits further investigation.

## 6. CONCLUSION

In this study, we used available summer temperature and salinity profiles in Baffin Bay and the Labrador Sea to estimate the
SFV and thickness in this area. We compared the climatological results of SFV with a freshwater budget established by combining reanalysis data for P-E, a mass balance model data for glacier ice melt, an energy balance model for river runoff, satellite imagery analysis for sea ice exported from the Arctic Ocean, and satellite-derived sea ice thickness data. Further, we compared SFV for climatological periods before and after 1995 to evaluate the effects of multidecadal climate changes on the summer freshwater input. Lastly, we compared SFV with total Freshwater Content (FWC) estimated following standard
procedures (e.g., Carmack et al.; 2008, Proshutinsky et al., 2019). The main conclusions of this study are as follows:

- The methodology for estimating summer freshwater thickness yields reliable results, allowing us to distinguish the effects of various freshwater sources on the distribution of freshwater thickness. With a climatological SFV of 1956 km$^3$ against 2286 km$^3$ for the freshwater budget, the two methods of estimate cross-validate each other.
- The summer freshwater budget for local freshwater sources is dominated by ice melting. Sea ice and glacier
melt/iceberg calving represent 37% and 26% of the summer freshwater volume, respectively.
- At the multidecadal scale, SFV exhibits an increase in Baffin Bay (+226 km$^3$) and a decrease in the Labrador Sea (-112 km$^3$). The changes in the Baffin Bay SFV are dominated by the increase in glacier melting and iceberg calving, whereas the SFV decrease in the Labrador is likely due to the decline in sea ice volume starting after the mid-1990s.




- At the interannual time scale, the variability of FWC in the Labrador Sea is caused primarily by that of the SFV in
this area. However, since the end of the 1990s, remote sources of freshwater, from either Hudson Bay or the Beaufort
         Sea might be becoming the dominant factors.

- The multidecadal variations of SFV and FWC are decoupled in the Labrador Sea. On the long-term trend, FWC has
      increased by nearly 1000 km$^3$ since 1950, whereas SFV has decreased by 140 km$^3$ (Fig. 12). We suggest that the
      increase in the Beaufort Gyre FWC since the 1970s and subsequent release into the Labrador Sea is responsible for
the long-term changes of FWC in the Labrador Sea.

Our results suggest that a rather rapid increase in FWC occurred in the Labrador Sea during the 1990s rather than a steady growth throughout the 70-year period of our study. This is consistent with studies suggesting that the release of freshwater from the Beaufort Gyre may be intermittent (e.g., Proshutinsky et al., 2019; Zhang et al., 2021). Previously, the release of freshwater from the Arctic Ocean generated the great salinity anomaly of the 1970s, mainly through export via the Fram Strait to the northeastern Atlantic Ocean (Belkin et al., 1998). Our results suggest that large freshwater anomalies from the Beaufort Sea could also reach Baffin Bay and the Labrador Sea via the Nares Strait and other straits of the Canadian Arctic Archipelago. This would be in addition to the increase in the volume of Arctic sea ice exported via the Nares Strait (Moore et al., 2021). As the FWC in the Beaufort Sea has been rising dramatically for the past two decades, the next period of freshwater release could further increase the FWC in the Labrador Sea. Since some of the surface water from the Labrador Current ultimately reaches the interior of the Labrador Sea (e.g., Fratantoni and McCartney, 2010; Jutras et al., 2023), such changes could have a major impact on the winter convection and formation of the Labrador Sea Water, and possibly the North Atlantic Deep Water. In the future, comparing the time series of the Beaufort Gyre freshwater content, atmospheric pressure above the Beaufort Sea, and the Labrador Sea freshwater content could clarify the influence of the Beaufort Gyre freshwater content variability on the Labrador Sea surface properties.




*Data availability.* The marine mammal data were collected and made freely available by the International MEOP Consortium and the national programs that contribute to it. (http://www.meop.net). Argo float data is collected and made freely available by the International Argo Program and the national programs that contribute to it. (https://argo.ucsd.edu, https://www.ocean-

ops.org). The Argo Program is part of the Global Ocean Observing System. Argo float data were downloaded together with other historical CTD data as part of the World Ocean Database 2018, a National Centers for Environmental Information standard product. The production of the merged CryoSat-SMOS sea ice thickness data was funded by the ESA project SMOS & CryoSat-2 Sea Ice Data Product Processing and Dissemination Service, and data from 2011 to 2022 were obtained from AWI. The dataset of proxy sea ice thickness by Glissenaar et al. (2023) is available from the following address

https://doi.org/10.5281/zenodo.7644084. The year-round, bi-monthly CryoSat-2 sea ice thickness dataset of Landy and Dawson (2022) is available from https://doi.org/10.5285/d8c66670-57ad-44fc-8fef-942a46734ecb.

*Author contributions.* VM and KO designed the experiments, and VM and MI carried them out. KF developed the models code and performed the simulations. SH designed and carried out the processing of SAR images. VM and MK designed the

ocean data processing and VM and MI carried it out. VM prepared the manuscript with contributions from all co-authors.

*Competing interests.* At least one of the (co-)authors is a member of the editorial board of The Cryosphere.

*Acknowledgements.* This work is supported by Grants-in-Aid for Scientific Research #22K1409402, #17H01157, and

#20H05707 and the Arctic Challenge for Sustainability (ArCS) and ArCS II project from the Ministry of Education, Culture, Sports, Science and Technology in Japan. This work was also supported by a research fund for Global Change Observation Mission Water 1 (GCOM-W1) of the Japan Aerospace Exploration Agency (JAXA) (PI No. ER2GWF404 and ER3AMF424). Hersbach, H. et al., (2023) was downloaded from the Copernicus Climate Change Service (C3S) (2023). The dataset used in this study contains modified Copernicus Climate Change Service information (2023). Neither the European Commission nor

ECMWF is responsible for any use that may be made of the Copernicus information or data it contains. Data analyses were conducted using the Pan-Okhotsk Information System of Hokkaido University and all manuscript figures were drawn using MATLAB version R22b (The MathWorks, Inc. https://uk.mathworks.com/products/matlab.html).



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
