# Peer review of "Estimation of ice melt, freshwater budget, and their multi-decadal trends in the Baffin Bay and Labrador Sea"

_EGUsphere, 2023_

## Referee Comment (RC2)

"Estimation of ice melt, freshwater budget, and their multi-decadal trends in the Baffin Bay and Labrador Sea" by Vigan Mensah, Koji Fujita, Stephen Howell, Miho Ikeda, Mizuki Komatsu, and Kay I. Ohshima deals with a very important matter freshwater budget of the Labrador Sea.

I really like the idea of the paper, support the importance of studying responses of changes in freshwater budget on ocean salinity, and the authors compiled published data from different, but not all external sources of freshwater to some up with general estimates to compare the changes with observed salinity changes. However, unfortunately, I cannot recommend this manuscript for publication, as the most important part of the paper – mapping of salinity and quantification of salinity changes has major problems and issues with data merging and synthesis and interpretation of temporal and spatial variability. There are obvious inconsistencies in the shown trends with those reported at specific locations before.

Apparently several different approaches brought together are required to extract seasonality and spatial gradients from the data, and last but not least account for changes in locations of sampling over the entire time period as well as different sampling method. The latter was done, like the authors tried to estimate how the values would change if a high resolution profile was subsampled in fewer points, but the reason for low-resolution in the 1960s (Nansen and Amundsen bottles) and presently (mammals and Argo float) are different, and so is the distribution of sampling depth over vertical profiles. I will explain a few issues, which are expected to change results uncontrollably. This might not happen, for example, is the same sampling grid was used in every year and all data collected evenly through summer season, and if the ice edge was located at the same position not affecting where data collected each year. Unfortunately, all of these factors change and the region is large to the point that the signal may have different arrival times and mixing rates affecting overall average conditions.

What I list before is a mix of major and minor comments, but I did not go for all minor suggestion, as the whole work needs to be redone, in my humble opinion:

The authors so not seem to be familiar with the recent publications on this subject (only a few are listed below, and more can be found by going from there).

What is "the upper layer of the North Atlantic Deep Water"?

I do not see any connection of this work with "the Atlantic Meridional Ocean Circulation". For example, this could be done by showing correlation with salinity in the Labrador Sea (e.g., [6, 8]).

"establish climatologies" – it is not a trivial task, and the author did not establish climatologies, they just blended all random data in certain time frames disrespectful of seasonal cycle. I can tell for sure that there was no some measurement near the glacier many years ago, whole new projects provided those, so there is a bias there. Same goes to the mammal data – they might travel through locations not sampled by the ships before, so there would be a bias. Overall, this means that each "climatology" is incomplete, biased by seasonal cycle (e.g., [8., 6]), and varying data density overlain on spatial gradients.

"assess the impact of multi-decadal climate change" – the impact on change on what?

"the influx of freshwater from the Beaufort Sea dominates the long-term variability" – this implies that the found freshwater signal does not follow the changes from all sources. Given that the oceanic

freshwater like have significant errors, the oceanic signal was probably unresolved. So, this stamen may not be accurate.

Line 25: I would recommend the authors to check more recent studies showing and discussing the variations in the Labrador Sea conditions and processes. Clarke and Gascard, 1983 – is a great work for its time, but it shows just one cruise. Smethie et al., 2000 is just one of many chemical tracer works. There were recent studies showing connection between temperature and salinity signals in the Labrador Sea and convection. Many previous works on freshwater transport were done by Bob Dickson (not mentioned once).

"preexisting weak stratification leads to deep convection events" – It is not a wrong statement, buy I would avoid it, as it remind me of a chicken and an egg - indeed, weak stratification is a result of convection – it would not be weak without convection.

"with the mixed layer reaching depths of up to 1600 m" – is this a mean or maximum depth? Neither is correct, by the way. The references below, show the whole range of convection.

"The properties of the surface waters and the associated stratification in the Labrador Sea play an important role in establishing deep convection events" – Again, the link is important, but missing in the paper. A study like this might benefit from including the de-seasoned salinity from the central Labrador Sea to show if there is any lagged or no-lagger correlation.

"and it is thus essential to document the temporal changes in these waters' properties." – again, this was an area of dense recent research (as shown in the references below).

"The cumulation of the freshwater discharge all along this anticlockwise circulation explains the cold and fresh properties of the surface waters in the Labrador Sea." – I disagree about the reason for cold in the Labrador Sea. Fresh water warms to 6C or higher at surface in summer.

"A considerable amount of temperature and salinity profiles have been acquired during the ice melting season (~May to September) since the 1950s, and using these data could be a good way to document long-term changes in the summer freshwater inputs in Baffin Bay and the Labrador Sea." – I disagree with the sufficiency of these data for proper documenting. Given the seasonal cycle and scatter over large area, the coverage is not adequate for straight-forward assessment of freshwater content changes. More sophisticated approaches needed than a gross merger of all data.
It ,may seem that 100-200 profiles are many, but spreading those over a long summer period with strong seasonal cycle and over a long shelf and across the shelf with strong gradient makes a few hundred really sparse, especially of mostly concentrated in a specific location … A thorough analysis of data distribution is needed.

"The main source of temperature and salinity data for this study is historical conductivity-temperature-depth (CTD) data" – CTD data were not present in early years in large quantities, and the early CTD data had poor quality. Could it be that NODC calls "CTD" data any vertically-interpolated bottle data profile?

"supplemented with data from the Marine Mammals Exploring the Ocean Pole to Pole (MEOP) database of CTD biologging observations" – I worked with the seal data before (although, never publishes), and found that, while being manageable for temperature (not always), salinity requires thorough checking, careful cleaning and calibration. It is doable, but needs some effort, looking for crossovers, etc.

"Data were selected from May to August in the Labrador Sea and from May to September in Baffin Bay, that is, up to one month after the end of sea ice melt in the respective regions." – Unfortunately, I strongly object this approach, and cannot agree anyone merging all profiles for the top 200 m over four month (there is still much ice in May, by the way). **The seasonal cycle contributes to all signals, especially on shelves.** I address the authors to [8], where I show seasonal cycles in salinity just for some years. I remove seasonal cycle at all locations and all depths by "smart" multi-parametric fitting. The seasonal coverage changed over years, and I know this for sure, and this alone gave a bias to the assessment by the authors. The same goes to regional position – there are strong horizontal gradients as well as differences between Canadian and Greenland shelves.

"A total of 47942 profiles are available, spanning from 1950 to 2022." – given the size of the region and seasonality, this number is not clear. How many mammals, how many real CTDs, uninterpolated bottles? I guess after cleaning this number drops to 7,000-10,000, which divided by all years and given the specific I described before may not be much.

"The temperature minimum in the upper 100 m should be less than 0°C. This criterion ensures that winter water still exists in the water column (Fig. 2)." – So, the authors basically select profiles that have a Cold Intermediate Layer (CIL)? Not clear why they needed it. Moving off the shelf CIL disappears, while freshwater is present. So, I say this step may create a bias, especially toward end-summer, when the major chunk of freshwater comes in.

Overall, the conditions 160-170 look artificial and impose certain biases on remaining data. I would carefully rethink the while strategy of data selection. Excluding profiles within certain limits of temperature and salinity may remove important anomalies or vice versa. For example, if you have a high salinity threshold, imagine having profiles right near that threshold, so this cluster will create a positive bias. I would also note that high ice cover in some years might strongly bias the results, as no data came from under the ice, so the missing parts of the domain …

"The other 5702 profiles were acquired by high-resolution CTD" – for what period?

There are some further issues with data selection and exclusion, but the seasonality and overall shift in observations from year to year create large errors.

"This small bias does not change the conclusions of this paper." The problem here is that there was no CTD in the 1950, and the number of points is often less than 10, however, depending on what was used as the NODC source, they might interpolate between bottle depths to provide CTD-like resolution). The other problem is that the low-resolution patterns were different in the first and second periods.

1996 was used for partitioning, but I do not think it was a good choice. Furthermore, the studies of the Greenland melt effect on salinity, like

https://agupubs.onlinelibrary.wiley.com/doi/full/10.1029/2018JC014686

https://journals.ametsoc.org/view/journals/clim/34/22/JCLI-D-20-0610.1.xml

showed that the effect is broadly distributed and hence small.

Figure 7 shows actual points used to construct the estimates, and there are some isolated high-value lines demonstrate my point about possible biases. Those could affect overall values for that year. Could

those points be from a mammal showing lower salinity because conductivity cell was too dirty? All of these cases need proper checking.

I can point at an obvious artifact of the method in Figure 8. What is seen there is a strong spatial differentiation of freshwater thickness change between two long periods. Of course, Greenland melt may be the factor of nearshore increase, but where there any data in that region before? Those might be spikes brought by recent projects in that area, as well as glacier retreat opening larger shelf area for sampling. The fact those were not seen before, does not mean those did not exist. The changes from positive to negative from north to south are also questionable, because, even if those are related to polynya, and any large-scale shift, the effect would be smeared over summer and blended between years – the currents are really string there, therefore, the authors should see more continuity on longer time scales. There are ways of dealing with the issues like this, but using different methods.

There is high freshwater areas in Figure 8b, and missing by the coast in 8a. These could result from the mentioned biases. Again, the periods are too long to expect such changes in overall position in freshwater clusters. The currents are strong and persistent. There were observations near glaciers recently and before, here are the inshore peaks in 8b. Even of Greenland melt was lower before, it was not as insignificant as in 8a.

I do not go over details of river, glacier and other discharges. I think these all are needed, and may be further tuned along the suggested lines, but it is critical to get the oceanic part properly and accurately addressed.

It is not clear what is the difference between SFV and FWC. Is the latter based on year-round data (again seasonality and spatial data shifts may be significant), or one is full-depth, while the other depth-restricted. A vertical partitioning of freshwater was suggested in [6], which may give a good point for linking and comparison, once the data issues are resolved.

The effect of the Beaufort Gyre release on salinity is yet to be seen – it only started about five yeas ago, and is expected to reach in about 7-10 year.

I wish the authors best of luck and a successful reworking the observational part.

[1] The "great salinity anomaly" in the Northern North Atlantic 1968–1982 - ScienceDirect

[2] A change in the freshwater balance of the Atlantic Ocean over the past four decades - PubMed (nih.gov)

[3] Recent changes in the North Atlantic | Philosophical Transactions of the Royal Society of London. Series A: Mathematical, Physical and Engineering Sciences (royalsocietypublishing.org)

[4] Changing freshwater content: Insights from the subpolar North Atlantic and new oceanographic challenges - ScienceDirect I would recommend the whole special issue

[5] Current estimates of freshwater flux through Arctic and subarctic seas - ScienceDirect

[6] A new collective view of oceanography of the Arctic and North Atlantic basins - ScienceDirect I would recommend the whole special issue

[7] Recurrent replenishment of Labrador Sea Water and associated decadal-scale variability - Yashayaev - 2016 - Journal of Geophysical Research: Oceans - Wiley Online Library

[8] Meteorological, Sea Ice and Oceanographic Conditions in the Labrador Sea during 2020 (dfo-mpo.gc.ca)